# *Lactobacillus reuteri* in Its Biofilm State Improves Protection from Experimental Necrotizing Enterocolitis

**DOI:** 10.3390/nu13030918

**Published:** 2021-03-12

**Authors:** Ameer Al-Hadidi, Jason Navarro, Steven D. Goodman, Michael T. Bailey, Gail E. Besner

**Affiliations:** 1Department of Pediatric Surgery, Nationwide Children’s Hospital, The Ohio State University College of Medicine, Center for Perinatal Research, The Research Institute at Nationwide Children’s Hospital, Nationwide Children’s Hospital, 700 Children’s Drive, Columbus, OH 43205, USA; ameer.al-hadidi@nationwidechildrens.org; 2Center for Microbial Pathogenesis, The Research Institute at Nationwide Children’s Hospital, 700 Children’s Drive, Columbus, OH 43205, USA; Jason.navarro@gmail.com (J.N.); Steven.goodman@nationwidechildrens.org (S.D.G.); Michael.bailey2@nationwidechildrens.org (M.T.B.)

**Keywords:** necrotizing enterocolitis, prematurity, *Lactobacillus reuteri*, probiotics, dextranomer microspheres

## Abstract

Necrotizing enterocolitis (NEC) is a devastating disease predominately found in premature infants that is associated with significant morbidity and mortality. Despite decades of research, medical management with broad spectrum antibiotics and bowel rest has remained relatively unchanged, with no significant improvement in patient outcomes. The etiology of NEC is multi-factorial; however, gastrointestinal dysbiosis plays a prominent role in a neonate’s vulnerability to and development of NEC. Probiotics have recently emerged as a new avenue for NEC therapy. However, current delivery methods are associated with potential limitations, including the need for at least daily administration in order to obtain any improvement in outcomes. We present a novel formulation of enterally delivered probiotics that addresses the current limitations. A single enteral dose of *Lactobacillus reuteri* delivered in a biofilm formulation increases probiotic survival in acidic gastric conditions, increases probiotic adherence to gastrointestinal epithelial cells, and reduces the incidence, severity, and neurocognitive sequelae of NEC in experimental models.

## 1. Introduction

Necrotizing enterocolitis (NEC) is a disease that has been a major source of morbidity and mortality for premature neonates for decades. Affecting 10% of infants with birth weight < 1500 g, NEC is a neonatal intestinal disease that is manifested by excessive inflammation that may progress to tissue destruction, bacterial translocation, and sepsis. The disease carries a mortality rate as high as 20–30% [1,2]. Despite decades of research and an estimated annual cost to the health care system of nearly USD 1 billion, NEC remains the number one cause of death from gastrointestinal disease in premature infants [2]. Thus far, treatment and attempts at prevention of NEC have remained subpar, with surviving infants often being left with debilitating morbidities including short-gut syndrome, cholestatic liver disease, and poor growth and neurodevelopmental outcomes [3,4].

The etiology of NEC is multi-factorial with prematurity, low birth weight, administration of enteral feeds, and antibiotic exposure associated with development of the disease [2,5]. Bacterial colonization of the gastrointestinal tract is essential to healthy gut development, with strong evidence indicating that gut dysbiosis plays a prominent role in patient vulnerability and development of NEC [6,7,8,9,10]. Large proportions of beneficial health-promoting bacteria, including *Lactobacillus* and *Bifidobacteria* species, are present in healthy full-term breast-fed infants [11,12,13]. Additionally, breast milk contains significant amounts of undigestible oligosaccharides that play a role as prebiotics, nurturing and promoting the growth of the favorable gut microorganisms necessary for bacterial-epithelial cross talk, which is crucial for nascent gut and immune system development [11,12]. Conversely, premature infants have reduced microbiome diversity and stability, with smaller proportions of beneficial bacteria including *Lactobacillus* and *Bifidobacterium* species, and increased levels of bacteria that can become pathogenic including Gammaproteobacteria (i.e., *Escherichia coli, Klebsiella pneumoniae*), which is evident in infants that develop NEC [10,14,15,16,17,18,19,20].

To counter the altered intestinal microbiome and to reduce the pathogenic bacterial colonization frequently seen in premature infants, administration of probiotics, or live microorganisms that confer a health benefit on the host, emerged as a means of NEC prevention in the late 1990s [21,22]. Since then, numerous trials evaluating the efficacy of probiotics in preventing NEC have been conducted, with some demonstrating favorable results [23,24]. Oral administration of *Lactobacillus* and *Bifidobacterium* was shown to prevent NEC in very low birth weight infants [25,26], and when administered in combination with breast milk, there was greater reduction in the incidence of NEC compared to infants receiving breast milk alone [6,27]. Furthermore, using animal models of experimental NEC, probiotics have been shown to inhibit inflammation, reduce apoptosis, inhibit Toll-like receptor 4 (TLR4) activation, and protect against intestinal mucosal barrier breakdown [28,29,30,31,32].

However, there are significant concerns and limitations regarding the current method of probiotic administration. An acidic gastric environment, interactions with bile acids, pressure from the host immune system, and competition with commensal and pathogenic bacteria can rapidly render probiotic bacteria ineffective, with a crippled capacity to adhere to and colonize the gut [33,34]. Due to the inability to be retained within the host, oral administration is required daily, if not multiple times per day, to witness even a modest beneficial effect that is effectively lost upon the cessation of probiotic administration [35]. Additionally, repeated administration of oral probiotic bacteria to premature infants with compromised gut barrier function can be problematic, given the risk of inducing bacteremia or sepsis from the probiotic administered [36,37,38,39].

To overcome these concerns and limitations, we introduced a novel probiotic delivery system that delivers beneficial health-promoting *Lactobacillus reuteri* (ATCC 23272) in a biofilm state rather than in a free-living planktonic state [40,41]. Probiotics delivered as a biofilm, i.e., an adhered or aggregated community of bacteria that produce a self-forming protective matrix of DNA, proteins, lipids, and oligosaccharides, are more resistant to harsh environmental conditions such as acidic gastric pH, laminar/turbulent fluid forces, anti-microbial agents, and host immune defenses compared to free-living planktonic bacteria [42,43]. The use of probiotics in their biofilm state has been investigated and utilized in a few conditions, including antagonizing pathogenic infections in implants and incorporation into anti-neoplastic strategies as immunoregulators [44,45,46,47]. However, the delivery of probiotics in a biofilm state is a new and innovative strategy in the management and prevention of NEC. This review describes our findings from multiple publications using our novel probiotic delivery system, as well as the potential future applications it may bring to neonatal care.

## 2. Main Body

### 2.1. Novel Delivery System

#### 2.1.1. *Lactobacillus reuteri*

*L. reuteri* is a Gram-positive bacterium that is naturally found in a variety of hosts and environments, including the healthy human intestine [48,49,50]. *L. reuteri* strains commonly found in humans are divided into two clades, i.e., clade II and clade IV, that are genetically and functionally distinct [51]. *L. reuteri* ATCC 23272 (aka, *L. reuteri* DSM20016) is a clade II strain originally isolated from human breast milk that is a frequently used probiotic given its anti-inflammatory and antimicrobial properties. Anti-inflammatory abilities are in part attributable to its ability to produce histamine and diacylglycerol kinase that together lead to activation of histamine H2, but inhibition of histamine H1 receptors (respectively) [52,53,54]. The H2 receptor is highly expressed in the intestine and has anti-inflammatory effects [55]. In addition to histamine, *L. reuteri* folate metabolism has been linked to its anti-inflammatory properties [56], including the production of ethionine which can modify human chromatin through ethylation [57]. *L. reuteri* has also been shown to induce anti-inflammatory T regulatory cells, suppress T helper (Th) 1 and Th2 cytokine responses, and to alter dendritic cell activity, but the mechanisms by which this occurs are not widely understood [58,59,60,61]. Its anti-microbial abilities are due to its production of the anti-microbial compound 3-hydroxyproprionaldehyde (3-HPA), also known as reuterin [61,62,63]. Reuterin is efficient at inhibiting the growth of a number of gastrointestinal pathogens through induction of oxidative stress [61,63,64,65]. Additionally, *L. reuteri* has extracellular glucosyltransferase (GTF) proteins that catalyze the formation of exopolysaccharides of glucose (glucans) from disaccharide sugars (e.g., maltose or sucrose) and possess glucan binding domains, that further allows for strong binding to other glucans.

#### 2.1.2. Production of *L. reuteri* Biofilm by Adherence to Dextranomer Microspheres

We use dextranomer microspheres (DMs) as a surface for *L. reuteri* attachment and biofilm formation [40]. DMs are porous, semi-permeable, biocompatible, biodegradable, non-immunogenic, non-allergic, Generally Recognized As Safe (GRAS) microspheres composed of cross-linked dextran. DMs are currently being used in several Food and Drug Administration (FDA)-approved medical products and are accepted as safe for human administration [66,67,68]. In order to create a probiotic biofilm, cultures of *L. reuteri* are introduced to DMs and undergo a brief incubation period to allow for adherence and biofilm formation (Figure 1). Importantly, known pathogens including *Escherichia coli*, *Salmonella typhimurium*, and *Clostridioides difficile* do not detectibly bind to DMs, thereby not providing pathobionts with a scaffold to adhere and grow [40]. Additionally, because DMs are porous, they can be preloaded with nutritious prebiotic substances that contribute to probiotic growth and promote further biofilm production. For example, disaccharides, that under regular circumstances would be promptly diluted, metabolized, and absorbed within the proximal gastrointestinal tract, will remain undiluted within the DMs and gradually diffuse out to provide their beneficial prebiotic contents at high concentrations discriminatively to the adhered probiotics. DMs are used to take advantage of *L. reuteri’s* GTF native ability to bind to cross-linked dextran. The GTF-dependent selective binding of *L. reuteri* to DMs results in a biofilm state with: (1) enhanced binding of *L. reuteri* to intestinal epithelial cells, (2) protection against low gastric pH, and (3) access to high concentrations of beneficial luminal substances to *L. reuteri* in order to augment its probiotic effects.

### 2.2. Effects of L. reuteri in Its Biofilm State In Vitro 

#### 2.2.1. *L. reuteri* Adherence to Dextranomer Microspheres Is Dependent on GTF In Vitro

Adherence of *L. reuteri* to DM is heavily GTF-dependent and is enhanced in the presence of certain disaccharides, e.g., maltose and sucrose (Table 1). Bacteria bound to DM are resistant to acidic conditions and have enhanced adherence to human intestinal epithelial cells [40]. In an effort to enhance *L. reuteri* adherence to DM, semi-permeable DMs were loaded with beneficial luminal cargo to promote improved probiotic adherence in vitro [40]. After differential staining of *L. reuteri* with a green fluorescent nucleic acid stain (SYTO9) and DMs with Congo Red, confocal laser scanning microscopy revealed significantly increased *L. reuteri* adherence to DMs containing sucrose or maltose compared to DMs containing water only [40].

#### 2.2.2. Beneficial Cargo Enhances *L. reuteri* Survival at Low pH In Vitro

A major hindrance to the efficacy of orally consumed probiotics is the acidic nature of the stomach [70]. This is problematic given the need for viable *L. reuteri* to reach the distal gastrointestinal tract to be effective in preventing NEC. We investigated whether *L. reuteri* bound to DM would have increased survival under acidic conditions, and whether the addition of beneficial luminal cargo would further enhance survival in a GTF-dependent manner.

The viability of *L. reuteri* was evaluated after placement in synthetic gastric acid with a pH of 2 for 4 h. *L. reuteri* in its planktonic state had a survival of <0.1%, with no significant increase in survival in the presence of sucrose or maltose, or when adhered to water-filled DMs. However, *L. reuteri* in its biofilm state adhered to DMs loaded with sucrose or maltose demonstrated significantly improved survival in the acidic environment. Notably, no significant improvement in survival was seen with a mutated *L. reuteri* strain lacking GTF, even in the presence of DM loaded with beneficial cargo [40]. These findings demonstrate the importance of GTF-dependent adherence of *L. reuteri* to DMs loaded with beneficial luminal cargo.

#### 2.2.3. Beneficial Cargo Enhance *L. reuteri* Adherence to Human Intestinal Epithelial Cells In Vitro

We next investigated whether *L. reuteri* in its biofilm state adhered to DM would improve adherence of the probiotic to human DLD-1 intestinal epithelial cells (adult human colonic epithelial cells) and to FHs 74 Int cells (3–4-month gestation small intestine epithelial cells) in vitro. We also examined whether enhanced adherence was promoted by the addition of beneficial luminal cargo within DM.

When comparing wild-type *L. reuteri* to the GTF mutant strain, we found significantly increased binding to DLD-1 cells with wild-type *L. reuteri*, regardless of whether *L. reuteri* was delivered in its planktonic or biofilm state. This signifies the importance of GTF in adherence of the probiotic to intestinal epithelial cells. When *L. reuteri* was bound to DMs containing sucrose or maltose, adherence to DLD-1 cells increased by 4.7-fold and 5.2-fold, respectively. Although there was lower probiotic adherence to FHs 74 Int small intestine epithelial cells compared to colonic cells, *L. reuteri* in its biofilm state bound to DMs loaded with sucrose or maltose demonstrated a 1.8-fold and 2.7-fold increase in adherence, respectively, compared to *L. reuteri* in its planktonic state [40].

### 2.3. Delivery of Lactobacillus reuteri in Its Biofilm State In Vivo 

#### 2.3.1. Murine Animal Model of Experimental Necrotizing Enterocolitis

Sprague Dawley rat pups are delivered prematurely from timed-pregnant dams on gestational day 21 via terminal cesarean delivery. After delivery, pups are randomized into experimental groups that received a single 100 μL enteral dose of *L. reuteri* (2 × 10^8^ CFU), either alone or with 0.5 mg of DM, or sterile water control treatment via gastric gavage [71]. Pups are subjected to a modification of the stress protocol initially described by Barlow et al. to induce experimental NEC [40,41,70]. Pups receive hypertonic formula via orogastric gavage five times daily, with exposure to hypoxia (~100% nitrogen) and hypothermia (4 °C) three times daily for 90 s and 10 min, respectively. Additionally, pups receive lipopolysaccharide (2 mg/kg) once via gastric gavage on the first day of life. Pups that develop clinical signs of NEC are euthanized with collection of intestinal tissue for histologic evaluation. After 96 h, any surviving pups are euthanized, and intestine collected. Breast-fed control pups are not exposed to experimental stresses, and are breast fed by surrogate dams. Hematoxylin and eosin (H&E) intestinal tissue sections are graded blindly by two independent observers using an established histologic NEC injury grading system [72].

#### 2.3.2. *Lactobacillus reuteri* in Its Biofilm State Protects the Intestines from Injury and Preserves Gut Barrier Function during Experimental NEC

We initially evaluated the efficacy of *L. reuteri* administered as a single dose shortly after delivery in preventing NEC in our experimental NEC model [41] (Table 1). Animals exposed to NEC were either untreated (receiving sterile water only) or treated with *L. reuteri* in its planktonic state (*L. reuteri* alone), DM alone, or *L. reuteri* in its biofilm state (*L. reuteri* +DM). The only significant decrease in the incidence of NEC was seen with pups receiving *L. reuteri* in biofilm state. In addition, differences in intestinal mucosal permeability were evaluated by quantifying systemic absorption of enterally administered fluorescein isothiocyanate (FITC) labeled dextran [31]. FITC-dextran (1500 mg/kg) was administered to pups 48-h following cesarean delivery. Four hours after administration, pups were euthanized and serum FITC-dextran levels analyzed. Increased levels of serum FITC-dextran are indicative of increased intestinal permeability and impaired gut barrier function. Pups exposed to NEC that were untreated (receiving sterile water only) had significantly increased intestinal permeability compared to breast-fed control pups. Only pups exposed to NEC that were treated with *L. reuteri* in its biofilm state demonstrated significant reduction in intestinal permeability, indicative of improved gut barrier function [41]. Thus, a single dose of enterally administered *L. reuteri* in its biofilm state significantly decreased the incidence of NEC and improved gut barrier function, in a murine model of experimental NEC.

#### 2.3.3. Enhancing *Lactobacillus reuteri* Biofilm Formation Increases Protection against Experimental NEC, Improves Survival, Preserves Gut Barrier Function, Decreases Proinflammatory Cytokine Production, and Preserves Eubiosis during Experimental NEC

Our in vitro studies showed that *L. reuteri* in its biofilm state bound to DMs loaded with sucrose or maltose demonstrated enhanced survival at low pH and greater adherence to intestinal epithelial cells (Table 1). Based on this, we hypothesized that enhanced biofilm formation would translate to a more pronounced decrease in NEC incidence and severity with improved survival of rat pups in our experimental NEC model [69]. When the *L. reuteri* biofilm formulation containing beneficial cargo was introduced to the model, there was a marked decrease in NEC incidence in the rat pups receiving the formulation [69]. Approximately 61% of pups receiving no treatment developed NEC, with no significant decrease in the incidence of NEC in pups receiving a single dose of *L. reuteri* in its planktonic state, or in pups receiving DM loaded with sucrose alone. As previously demonstrated, pups receiving *L. reuteri* in its biofilm state bound to DMs without beneficial luminal cargo had a significant decrease in the incidence of NEC to 33%. Furthermore, pups receiving *L. reuteri* in its biofilm state bound to DMs containing sucrose or maltose, where biofilm formation is further enhanced, demonstrated substantial further improvement in protection against NEC, with NEC incidences decreased to 14% and 15%, respectively (*p* < 0.05). This improvement translated to enhanced survival, with nearly 60% of rat pups treated with *L. reuteri* in its biofilm state bound to DMs containing luminal sucrose or maltose surviving until the end of the 96-h protocol, whereas only 20% of untreated stressed pups survived [69]. No significant improvement in survival was appreciated in pups receiving *L. reuteri* in its planktonic form, or in pups receiving DM loaded with sucrose or maltose alone. Furthermore, any significant protective effect seen with *L. reuteri* was absent when a GtfW mutated strain was administered, even in the presence of DMs loaded with beneficial cargo.

To investigate gut barrier function, 48 h after initiating the experimental NEC protocol, rat pups received enteral FITC-dextran. Four hours after administration, pups were euthanized and serum FD70 levels analyzed. Untreated pups experienced significantly increased intestinal permeability compared to breast-fed control pups (*p* = 0.001) [69]. Pups treated with a single dose of *L. reuteri* in its biofilm state bound to DMs containing sucrose or maltose had significantly reduced intestinal permeability compared to untreated pups (*p* = 0.009 and 0.006, respectively), indicating improved gut barrier function. Pups treated with a single dose *L. reuteri* in its planktonic state did not demonstrate improved gut barrier function.

We next examined the expression of inflammatory cytokines interleukin (IL)-6, IL-1B, chemokine ligand (CCL)-2, CXCL-1, and IL-10 in intestinal specimens from the different treatment groups. Pups exposed to NEC receiving no treatment demonstrated significant elevation in the expression of IL-6, IL-1B, CCL-2, CXCL-1, and IL-10 in the small intestine compared to unstressed breast-fed rat pups (*p* < 0.002) [69]. Pups receiving a single treatment dose of *L. reuteri* in its biofilm state bound to DMs containing sucrose or maltose had significantly lower expression of IL-6, IL-1B, CCL-2, CXCL-1, and IL-10 compared to untreated pups, or to pups treated with *L. reuteri* in its planktonic form (*p* < 0.05). The decrease in production of pro-inflammatory compounds is likely secondary to the intrinsic ability of *L. reuteri* to produce histamine, which suppresses pro-inflammatory cytokine production, in synergy with a biofilm state allowing for an enhanced anti-inflammatory effect. Further investigation is required as discussed in Section 2.4.

The gut microbiota community structure and taxa composition of pups treated with *L. reuteri* in its biofilm compared to its planktonic state were investigated using 16S rRNA gene sequencing analysis [69]. Unweighted UniFrac analysis demonstrated distinctive clustering between unstressed pups, stressed pups treated with *L. reuteri* in its biofilm state bound to DMs containing maltose, and stressed pups treated with *L. reuteri* in its planktonic state. Notably, the microbiota community arrangement of pups treated with *L. reuteri* in its biofilm state clustered more closely with unstressed, vaginally delivered, breast-fed pups. Furthermore, taxa-level analysis demonstrated that *Lactobacillus* species abundance was more effectively maintained and closely related to unstressed, breast-fed controls in pups treated with *L. reuteri* in its biofilm state compared to its planktonic state (*p* < 0.05). In addition, *Lactobacillus* species abundance was inversely associated with NEC injury (Pearson r= −0.480, *p* = 0.01). Both the biofilm and planktonic formulations of *L. reuteri* effectively limited the abundance of *Enterobacter* species, a potential enteric pathogen, compared to unstressed pups and untreated stressed pups (*p* < 0.05).

#### 2.3.4. Decreasing *L. reuteri* Biofilm Production Decreases Protection from NEC

The ability of *L. reuteri* to adhere and form a biofilm is GTF-dependent and essential to provide intestinal protection from NEC. An experiment comparing the ability of wild type *L. reuteri* versus GTF-deficient *L. reuteri* in preventing NEC was conducted. Compared to untreated stressed pups, pups receiving *L. reuteri* +DM loaded with maltose had a significantly lower incidence of NEC (65% vs. 22%, *p* < 0.05) [69]. However, pups receiving mutant *L. reuteri* deficient in GTF (blunted biofilm forming capacity), even in combination with DMs, failed to demonstrate any significant decrease in NEC incidence compared to untreated pups (*p* > 0.05), signifying the importance of GTF-dependent biofilm formation in the prevention of NEC.

### 2.4. Future Investigations of L. reuteri to Better Understand Protective Properties and Long-Term Outcomes

#### 2.4.1. Utilization of *L. reuteri* Mutants to Better Understand the Properties of *L. reuteri* That Lead to Intestinal Protection from NEC In Vivo

*L. reuteri* possesses anti-inflammatory and antimicrobial properties which we suspect play an integral role in intestinal protection from NEC. *L. reuteri*’s ability to produce histamine from L-histidine is believed to play a crucial role in anti-inflammation, as histamine suppresses tissue necrosis factor (TNF) production [53,61,73,74,75]. Future experiments will include: (1) administered of *L. reuteri* with DMs loaded with L-histidine (to augment anti-inflammatory production) and (2) utilizing a histamine-deficient mutant of *L. reuteri (*to attenuate anti-inflammatory production). Both experiments will be investigated in an animal model to evaluate the effect that increased or decreased histamine production has on intestinal protection from NEC.

*L. reuteri* also possesses antimicrobial properties due to its ability to produce reuterin [61,64]. Reuterin has been shown to effectively inhibit pathogenic bacterial growth through the induction of oxidative stress [64]. Future experiments will include utilizing a reuterin-deficient mutant of *L. reuteri* to evaluate whether intestinal protection from NEC is attenuated in the absence of reuterin.

#### 2.4.2. *Lactobacillus reuteri* in Its Biofilm State Has Potential to Attenuate Neurocognitive Injury after Experimental NEC

Over 40% of NEC survivors are left with debilitating life-long neurocognitive and developmental impairments [76,77]. These include development of cerebral palsy, or other cognitive, psychomotor, auditory, and visual disabilities that result in lower cognitive ability, educational achievement, and inferior mental health compared to age-matched NICU patients without NEC [78,79,80,81]. It is well recognized that premature infants suffering from intestinal injury can have abnormal neurodevelopment; however, the etiology remains unclear and likely multifactorial [78]. Associations have been made between gut microbiota and brain function, with dysbiosis in preterm infants after NEC and/or sepsis being linked to poor neurologic outcomes [78,79,80,81,82]. Multiple routes of communication exist for bidirectional gut-to-brain communication (neuronal, immune, metabolic and endocrine) and they all physiologically intersect with the gut microbiome. Thus, treatments that target the gut microbiome may improve cognitive disorders while addressing underlying intestinal injury.

There is increasing evidence suggesting that decreased myelination is largely responsible for the neurodevelopmental sequelae and white matter abnormalities in survivors of NEC [83,84,85,86]. Microglia are the macrophages of the brain and have a role in immune defense and brain homeostasis [87,88]. The activation of microglia triggered by proinflammatory cytokines results in loss of oligodendrocyte precursor cells accountable for neuronal myelination, leading to diminished myelination with cognitive impairment following NEC [83,84,85,86]. As mentioned above, we previously demonstrated the ability *L. reuteri* in its biofilm state to down-regulate the expression of proinflammatory cytokines and up-regulate the expression of anti-inflammatory compounds such as histamine [69]. Future investigations will explore whether a single enterally administered dose of *L. reuteri* in its biofilm state can attenuate microglial activation, preserve neuronal myelination, and protect against neurocognitive injury in rat pups surviving our experimental NEC model.

## 3. Conclusions

Necrotizing enterocolitis continues to be a major source of morbidity and mortality for premature infants. Despite years of research and advancements in critical care, improvement in the outcomes of infants suffering from NEC are subtle at best. Probiotics have shown promise as a potential treatment to reduce the incidence and severity of NEC; however, current delivery methods present legitimate concerns. Our proposed method of delivering a single enteral dose of the probiotic *Lactobacillus reuteri* in a biofilm formulation alleviates most of these concerns. Investigations using our experimental animal model have demonstrated the ability of our *L. reuteri* biofilm formulation to significantly reduce the incidence and severity of NEC, decrease NEC-related mortality, stabilize the intestinal mucosal barrier, and down-regulate the production of proinflammatory cytokines. Given the effects of NEC and of gut microbes on infant neurodevelopment, future studies will determine whether our enhanced probiotic formulation will help prevent the deleterious effects of NEC on neurocognitive development.

Since probiotics offer a potential benefit in other infectious or inflammatory conditions, additional investigation is underway for the use of our enhanced probiotic formulation in the treatment and management of several other gastrointestinal diseases, including *Clostridioides difficile* colitis [89] and inflammatory bowel disease.

## Figures and Tables

**Figure 1 nutrients-13-00918-f001:**
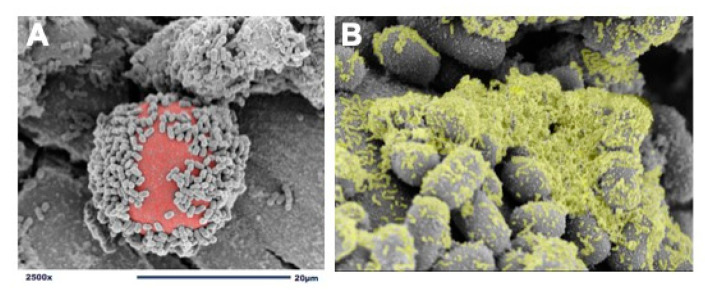
Adherence of *L. reuteri* to dextranomer microspheres. (**A**) scanning electron microscopy (SEM) image demonstrating the adherence of *L. reuteri* to the surface of a biocompatible dextranomer microsphere (DM; red); (**B**) magnified SEM image demonstrating the production of biofilm (green) by *L. reuteri* adhered to a sucrose-loaded DM.

**Table 1 nutrients-13-00918-t001:** Summary of in vitro and in vivo investigations with main results.

Article	Type of Study	Probiotic Strain	Aim of Study	Main Results
Olson et al. 2016 [41]	In vivo	*L. reuteri* ATCC 23272	To evaluate the efficacy of a novel probiotic delivery system in an experimental model of necrotizing enterocolitis	A single dose of *L. reuteri* in its biofilm state significantly decreased: (1) the incidence of NEC (2) the severity of NEC (3) intestinal mucosal permeability in premature rat pups, compared to free-living, planktonic *L. reuteri*
Navarro et al. 2017 [40]	In vitro	*L. reuteri* ATCC 23272	To evaluate the effect of enhancing a novel *L. reuteri* biofilm formulation with the addition of beneficial compounds as diffusible cargo within DMs	An enhanced probiotic formulation resulted in increased: (1) adherence of *L. reuteri* to DMs (2) resistance to acidic conditions (3) adherence to human intestinal epithelial cells in vitro
Olson et al. 2018 [69]	In vivo	*L. reuteri* ATCC 23272	To evaluate the efficacy of an enhanced novel *L. reuteri* biofilm formulation with beneficial cargo on protection from experimental NEC	A single dose of enhanced *L. reuteri* biofilm formulation with beneficial DM cargo resulted in decreased: (1) NEC incidence and severity (2) mortality (3) intestinal permeability (4) intestinal inflammation (5) alteration in gut microbiome compared to pups receiving *L. reuteri* biofilm formulation without beneficial DM cargo

## Data Availability

Data is contained within the article or supplementary material. The data presented in this manuscript are available in Olson et al. 2016 [41], Navarro et al. 2017 [40], and Olson et al. 2018 [69].

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
