# Peer review of "Lactobacillus reuteri in Its Biofilm State Improves Protection from Experimental Necrotizing Enterocolitis"

_nutrients, 2021, doi:10.3390/nu13030918_

Round 1

Reviewer 1 Report

Thank you for inviting me to review this review titled “Lactobacillus reuteri in its biofilm state improves protection from experimental necrotizing enterocolitis”. The authors in this review discuss a novel formulation of enteral Lactobacillus reuteri in a biofilm formulation compared to regular formulation in a planktonic form an experimental NEC model. 

  1. The major concern about this review is the way it was written. It is unclear from the abstract or introduction that you’re reading a review article. Instead I was under the impressions that it is a scientific paper. For example, the authors go at great lengths describing the model for NEC and the results of their experiments, but not much is written about other studies about the topic. I suggest the authors modify the title, abstract and body to reflect the nature of the manuscript. 
  2. Related to the comment above, some of the details listed in the body probably can be deleted as it pertains to the study the authors keep on referring to. Example is lines 230-238. 
  3. Finally, the goal of review articles should be to pick a topic, summarize the available literature, and come up with a conclusion with potential future needs. This review article fails to do so and only lists the rationale for using the novel formulation without discussion of any other methods or available literature on the topic. 

Reviewer 2 Report

In this manuscript, Ameer Al-Hadidi and colleagues reviewed the progress on investigating the effect of delivering Lactobacillus reuteri in a biofilm formulation on bacterial strain survival and its protection against experimental necrotizing enterocolitis.

The review is clearly designed and well-written which presents interesting findings. I only have a few issues that need to be addressed for further improvement.

Specifically,

  1. Please provide the lineage/strain information about the L. reuteri discussed in the review. In addition, the strain-specific immunomodulation of L. reuteri has been demonstrated, which could be included in the discussion section.
  2. In section 2.3.3, the discussion about the expression of inflammatory cytokines was limited. Please add interpretation of cytokine results.
  3. In the experimental necrotizing enterocolitis studies, please include results and discussions with respect to the impact of Lactobacillus reuteri+DM on microbial communities and colonization resistance to pathogens.

Minor issues,

1. L 50-51 – Commensal E. coli strains could be beneficial to the colonization of the gut microbiome in early life. The term of pathogenic bacteria may not be accurate.

2. L 62 – Please provide the full name of TLR. Please check through the manuscript regarding the abbreviation (such as GRAS, FDA, SEM, SYTO9, and cytokines). Please keep consistency with the format of Lactobacillus reuteri (L. reuteri vs. Lr).  

3. L 92 – “and has glucan binding domains”.

4. L 98-99 – Please revise the statement.

5. L 101 – “E. coli.

6. L 130 – “can drop as low as 1.5”.

7. L 245 – “proinflammatory TNF production”.

8. L 281 – “NEC remains a major source”.

9. L 295 – “C. difficile”.

Reviewer 3 Report

I read with interest this relevant review on a novel probiotic delivery system
that delivers beneficial health-promoting Lactobacillus reuter in NEC. This review present innovative data and it is well developed. Authors should be congratulated for the completeness of their review which includes a huge number of old and more recent studies. However, I suggest to try to be as concise as possible, to increase the readability of this paper. To this concern, the Authors could insert two Tables with the relevant paper: 

-in Vitro Study (authors - type of article - group of study (cell line ....)- aim of study - main results)

-in Vivo Study (authors - type of article - group of study (NEC model..)- aim of study - main results)

Round 2

Reviewer 1 Report

I have no other concerns.

Author Response

Thank you for your review. Responses were provided for all comments and suggestions.